# A Bibenzyl Component Moscatilin Mitigates Glycation-Mediated Damages in an SH-SY5Y Cell Model of Neurodegenerative Diseases through AMPK Activation and RAGE/NF-*κ*B Pathway Suppression

**DOI:** 10.3390/molecules25194574

**Published:** 2020-10-07

**Authors:** Mei Chou Lai, Wayne Young Liu, Shorong-Shii Liou, I-Min Liu

**Affiliations:** 1Department of Pharmacy and Master Program, Collage of Pharmacy and Health Care, Tajen University, Pingtung County 90741, Taiwan; meei@tajen.edu.tw (M.C.L.); ssliou@tajen.edu.tw (S.-S.L.); 2Department of Urology, Jen-Ai Hospital, Taichung 41265, Taiwan; waynedoctor@gmail.com; 3Center for Basic Medical Science, Collage of Health Science, Central Taiwan University of Science and Technology, Taichung City 406053, Taiwan

**Keywords:** AMP-activated protein kinase, neurodegenerative diseases, advanced glycation end-products, moscatilin, SH-SY5Y cells

## Abstract

Moscatilin can protect rat pheochromocytoma cells against methylglyoxal-induced damage. Elimination of the effect of advanced glycation end-products (AGEs) but activation of AMP-activated protein kinase (AMPK) are the potential therapeutic targets for the neurodegenerative diseases. Our study aimed to clarify AMPK signaling’s role in the beneficial effects of moscatilin on the diabetic/hyperglycemia-associated neurodegenerative disorders. AGEs-induced injury in SH-SY5Y cells was used as an in vitro neurodegenerative model. AGEs stimulation resulted in cellular viability loss and reactive oxygen species production, and mitochondrial membrane potential collapse. It was observed that the cleaved forms of caspase-9, caspase-3, and poly (ADP-ribose) polymerase increased in SH-SY5Y cells following AGEs exposure. AGEs decreased Bcl-2 but increased Bax and p53 expression and nuclear factor kappa-B activation in SH-SY5Y cells. AGEs also attenuated the phosphorylation level of AMPK. These AGEs-induced detrimental effects were ameliorated by moscatilin, which was similar to the actions of metformin. Compound C, an inhibitor of AMPK, abolished the beneficial effects of moscatilin on the regulation of SH-SY5Y cells’ function, indicating the involvement of AMPK. In conclusion, moscatilin offers a promising therapeutic strategy to reduce the neurotoxicity or AMPK dysfunction of AGEs. It provides a potential beneficial effect with AGEs-related neurodegenerative diseases.

## 1. Introduction

Many neurodegenerative diseases occur when nerve cells from the brain or peripheral nervous system are lost, ultimately leading to either functional loss (ataxia) or sensory deficits (dementia) [1]. The pathogenesis mechanism of neurodegenerative diseases is exceptionally complex. Oxidative stress due to the excessive production and release of reactive oxygen species (ROS) with a decreased level of detoxifying enzymes and antioxidant defenses has proposed a general pathological mechanism of major chronic neurodegenerative diseases [2]. A shred of growing epidemiological evidence has suggested that diabetic subjects are at increased risk of cognitive decline and the development of dementia [3]. The pathology associated with neurodegeneration involves chronic hyperglycemia, accelerating advanced glycation end products’ (AGEs) formation [4]. AGEs can then interact with the cell surface receptors (RAGE) leading to the elevation of cytosolic ROS, which causes nuclear factor-κB (NF-κB) nuclear translocation and cell apoptosis [5]. Apoptosis is driven by the disruption in the balance between the pro-apoptotic protein and the antiapoptotic protein, resulting in an elevation of mitochondrial permeability and parallel with a reduction in mitochondrial transmembrane potential (MMP), thereby, with a concomitant release of mitochondrial protein cytochrome c, leading to caspases’ activation [6]. Inhibition of the AGE-RAGE signaling axis is now considered promising in the prevention of diabetes/hyperglycemia-mediated neurodegenerative diseases [7].

An increasing number of studies have reported novel therapeutic interventions for neurodegenerative diseases. It has been documented that AMP-activated protein kinase (AMPK) activator metformin produces the neuroprotective effect during AGEs-mediated oxidative stress in human brain cells, suggesting that AMPK could be a critical therapeutic target against AGEs-mediated neuron damages [8]. AMPK is generally known as a serine/threonine kinase, which controls cellular energy levels by balancing energy requirements and nutrient usage via regulation of proteins involved in glucose and lipid metabolism [9]. AMPK is also expressed in neurons throughout the brain, where it might be involved in brain development, neuronal polarization, and neuronal activity [10]. Thus, blocked AGEs-RAGE signals and activation of AMPK are emerging as potential therapeutic targets for these neurodegenerative diseases [10,11].

There is growing evidence that many medicinal plants and natural compounds have potential adjunctive therapeutic effects on various neurodegenerative diseases [12]. Moscatilin (4-[2-(4-Hydroxy-3-methoxyphenyl)ethyl]-2,6-dimethoxy-phenol) is one of the active compounds from *Dendrobium* species, which was traditionally used as an immunomodulatory remedy against various diseases [13]. Besides, to show anticancer activity against many kinds of cancers [14,15], moscatilin has multiple pharmacological effects involving anti-inflammatory, antioxidant, and antiplatelet aggregation [16,17,18]. Moscatilin at concentrations equal to or less than 1 μmol/L has been reported to protect retinal cells from ischemia/hypoxia-induced damage and counteract methylglyoxal (MGO)-triggered oxidative damage and cell death in rat pheochromocytoma cells [19,20]. MGO is a dicarbonyl compound produced as a side product during glycolysis and a potent precursor of AGEs [21]. Thus, moscatilin might have a potential protective or restorative action on glycation that causes various neurodegenerative diseases, while the AMPK-dependent mechanisms mediating the beneficial effect of moscatilin remain to be fully elucidated.

The SH-SY5Y-derived neurons have become a popular cell model for investigating neurodegenerative diseases [22]. SH-SY5Y cells have also been utilized in neuropathy models, showing the effects of high glucose and AGEs [23]. In the present study, AGEs-induced injury in SH-SY5Y neuronal cells was used as an in vitro diabetic/hyperglycemia-associated neurodegenerative disease. Metformin was used as a positive control to clarify the role of AMPK signaling in the beneficial effects of moscatilin on the neurodegenerative diseases augmented under conditions of glycation.

## 2. Results

### 2.1. Influences on the Decreased Cell Viability in AGEs-Treated SH-SY5Y Cells

The cell viability was reduced from 85.3% to 58.6%, respectively, when SH-SY5Y cells were incubated with AGEs at concentrations from 5 to 100 μg/mL for 24 h (Figure 1A). A total of 50 μg/mL AGEs reduced the cell viability in SH-SY5Y cells in a time-dependent tendency (Figure 1B). Exposure of SH-SY5Y cells to 50 μg/mL AGEs for 24 h used to induce cell injury in the following experiments.

Moscatilin significantly reversed the decreased cell viability in AGEs-treated SH-SY5Y cells in a concentration-dependent manner (0.1–1 μmol/L) when cells were pretreated with moscatilin for 24 h before AGEs (50 μg/mL) exposure (Figure 1C). Pretreatment moscatilin (1 μmol/L) or metformin (2 μmol/L) sustains the survival rate of the AGEs-treated SH-SY5Y cells to 86.7% and 90.3%, respectively (Figure 1C). Compound C (5 μmol/L), in contrast, blocked the protective effect of moscatilin (1 μmol/L) or metformin (2 μmol/L) on AGEs-induced cell death (Figure 1D). Neither moscatilin nor metformin and Compound C alone affects SH-SY5Y cells (Figure 1D).

### 2.2. Changes in RAGE Expression and Oxidative Stress-Related Factors Induced by AGEs in SH-SY5Y Cells

The RAGE expressed in AGEs cultured SH-SY5Y cells was 4.4-fold higher than in untreated controls (Figure 2A). Pretreatment SH-SY5Y cells with moscatilin (1 μmol/L) or metformin (2 μmol/L) reduced the AGEs-induced upregulation of RAGE expression 52.6% or 57.8%, respectively, compared to the levels seen in the untreated controls. Compound C (5 μmol/L) pretreatment abolished the effects of moscatilin and metformin (Figure 2A).

AGEs significantly induced ROS generation, while the pretreatment of SH-SY5Y cells with moscatilin (1 μmol/L) or metformin (2 μmol/L) markedly inhibited AGEs-induced ROS generation (Figure 2B). Compound C (5 μmol/L) pretreatment reversed moscatilin or metformin against AGEs-induced ROS production (Figure 2B).

The intracellular levels of malondialdehyde (MDA) in SH-SY5Y cells cultured in AGEs medium were higher by 4.3-fold than the regular vehicle-treated group (Figure 2C). Pretreatment SH-SY5Y cells with moscatilin (1 μmol/L) or metformin (2 μmol/L) resulted in a decrease in MDA levels by 39.4% and 44.1%, respectively, in AGEs-incubated SH-SY5Y cells relative to those observed in the vehicle-treated counterparts (Figure 2C). Compound C pretreatment restored moscatilin and metformin’s actions on the reduction in the intracellular MDA levels in AGEs-treated SH-SY5Y cells (Figure 2C).

It observed that the lower GSH/GSSG ratio in SH-SY5Y cells exposed to AGEs, as well as moscatilin (1 μmol/L) or metformin (2 μmol/L), reversed this declined GSH/GSSG ratio (Figure 2D). Compound C obstructed moscatilin or metformin on the elevation GSH/GSSG ratio in AGEs-treated SH-SY5Y cells (Figure 2D).

### 2.3. Influences on Mitochondrial Functions in AGEs-Treated SH-SY5Y Cells

In AGEs-treated SH-SY5Y cells, the MMP was reduced by 56.8% of that in the regular group (Figure 3A). Moscatilin (1 μmol/L) or metformin (2 μmol/L) reduced the disruption of MMP in AGEs-cultured SH-SY5Y cells, but the addition of Compound C abolished the protective effects (Figure 3A).

The ADP/ATP ratio markedly increased when SH-SY5Y cells were incubated with 50 μg/mL AGEs for 24 h (Figure 3B). The treatment of SH-SY5Y cells with moscatilin (1 μmol/L) decreased the elevation of the ADP/ATP ratio induced by AGEs; it showed a similar result in the metformin (2 μmol/L)-treated group (Figure 3B). Compound C abrogated both the effects of moscatilin and metformin on the reversion of the AGEs, which induced a high ADP/ATP ratio (Figure 3B).

Increased cytosolic levels paralleled with the decreased mitochondrial cytochrome c in AGEs-cultured SH-SY5Y cells were observed (Figure 3C). Similar to the effect produced by metformin (2 μmol/L), the pretreatment of SH-SY5Y cells with moscatilin (1 μmol/L) inhibited the release of cytochrome c into the cytoplasm from the mitochondrial fractions induced by AGEs (Figure 3C). Compound C abolished the action of moscatilin (1 μmol/L) or metformin (2 μmol/L) on the reduction in mitochondrial cytochrome c release (Figure 3C).

There was a 5.3-fold elevation in the extent of apoptotic DNA fragmentation in the AGEs-cultured SH-SY5Y cells relative to the regular group. Following pretreatment with moscatilin (1 μmol/L) or metformin (2 μmol/L), the extent of apoptotic DNA fragmentation decreased by 42.3% and 50.6% in AGEs-cultured SH-SY5Y cells relative to the vehicle-treated counterpart group. In comparison, compound C lost the effect of moscatilin (1 μmol/L) or metformin (2 μmol/L) (Figure 3D).

### 2.4. Effects on the Pro-Apoptotic and Anti-Apoptotic Proteins in AGEs-Treated SH-SY5Y Cells

The protein level of p53 in SH-SY5Y cells under AGEs stimulated was elevated to 4.5-fold of the regular group (Figure 4A). Moscatilin (1 μmol/L) or metformin (2 μmol/L) pretreatment lowered the p53 level in AGEs-cultured SH-SY5Y cells by 39.8% and 47.3% of the vehicle-treated counterpart group, while both the effects of moscatilin and metformin were lost in the presence of Compound C (Figure 4A).

AGEs lowed the Bcl-2/Bax ratio to 12.3% in SH-SY5Y cells compared to that seen in the regular group (Figure 4A). The downregulation in the Bcl-2/Bax ratio observed in AGEs-cultured SH-SY5Y cells was significantly increased by pretreatment with moscatilin (1 μmol/L) or metformin (2 μmol/L) to 5.3- and 5.8-fold, respectively, relative to that of the vehicle-treated counterpart group (Figure 4A). Compound C reversed the action of moscatilin (1 μmol/L) or metformin (2 μmol/L) in SH-SY5Y cells under AGEs stimulation (Figure 4A).

The cleaved caspase-9 was 3.3-fold higher in the AGEs-cultured SH-SY5Y cells than in the regular medium cultured group (Figure 4B). In the presence of moscatilin (1 μmol/L) or metformin (2 μmol/L), cleaved caspase-9 levels in AGEs-cultured SH-SY5Y cells downregulated to 55.1% and 47.2% of those observed on the vehicle-treated counterparts (Figure 4B). Compound C blocked the action of moscatilin or metformin on the regulation of cleaved caspase-9 expression (Figure 4B).

AGEs caused a 4.2-fold increase in cleaved caspase-3 protein expression in SH-SY5Y cells (Figure 4B). The protein expression of cleaved caspase-3 in AGEs-cultured SH-SY5Y cells was markedly decreased (43.9% and 51.8% reduction, respectively) by pretreatment with moscatilin (1.0 μmol/L) or metformin (2 μmol/L) when compared to those of the vehicle-treated counterparts (Figure 4B). The reduction in cleaved caspase-3 protein expression in AGEs-cultured SH-SY5Y cells by moscatilin or metformin was abrogated in the presence of Compound C (Figure 4B).

The protein level of cleaved PARP in AGEs-cultured SH-SY5Y cells was higher by 3.5-fold of those in the vehicle control group (Figure 4B). Following the pretreatment of SH-SY5Y cells with moscatilin (1 μmol/L) or metformin (2 μmol/L), the higher PARP protein levels caused by AGEs were lowered to 51.4% and 34.2%, respectively; both the actions of moscatilin and metformin were blocked by Compound C (Figure 4B).

### 2.5. Effects on the AMPK and NF-κB Signaling in AGEs-Treated SH-SY5Y Cells

AGEs markedly decreased the p-AMPKα/AMPKα ratio in SH-SY5Y cells, which was about 32.6%, as seen in the normal group (Figure 5A). This AGEs-induced downregulation in the p-AMPKα/AMPKα ratio was significantly increased (2.0- and 2.1-fold elevation, respectively), by treatment with moscatilin (1 μmol/L) or metformin (2 μmol/L) when compared to that of their vehicle-treated counterpart group (Figure 5A). Compound C suppressed the effects of moscatilin or metformin on the elevation of the p-AMPKα/AMPKα ratio in AGEs-cultured SH-SY5Y cells (Figure 5A). Neither moscatilin nor metformin and Compound C changes the AMPKα protein levels in AGEs-cultured SH-SY5Y cells (Figure 5A).

The ratio of p-IκBα/IκBα in the cytosolic extract of AGEs-cultured SH-SY5Y cells was 37.8% lower than those in the vehicle control group, which was higher by 2.2- and 2.4-fold, respectively, in moscatilin (1 μmol/L) or metformin (2 μmol/L) pretreated group (Figure 5B). Compound C suppressed the action of moscatilin or metformin on the upregulation of cytosolic p-IκBα/IκBα (Figure 5B). The protein level of IκBα was not influenced by moscatilin or metformin and Compound C in AGEs-cultured SH-SY5Y cells (Figure 5B).

The ratio of nucleus p-p65/p65 was 4.3-fold higher in AGEs-cultured SH-SY5Y cells than in the vehicle control group (Figure 5C). The pretreatment of AGEs-cultured SH-SY5Y cells with moscatilin (1 μmol/L) or metformin (2 μmol/L) lowered the ratio of nucleus p-p65/p65 by 61.5% and 51.5%, respectively, relative to that seen in their untreated counterparts (Figure 5C). The protein expression of p65 was not altered by moscatilin or metformin in AGEs-cultured SH-SY5Y cells (Figure 5C). Compound C did not change the p65 protein levels in AGEs-cultured SH-SY5Y cells but blocked the downregulation of moscatilin (1 μmol/L) or metformin (2 μmol/L) on the nucleus p-p65/p65 ratio in AGEs-cultured SH-SY5Y cells to 3.5- and 3.6-fold, respectively, of that in the vehicle control group (Figure 5C).

## 3. Discussion

It has confirmed that AGEs’ interactions with their receptors (RAGE) play a role in the pathogenesis of diabetic complications and neurodegenerative disorders [4]. Anti-glycating systems, such as those preventing AGEs formation and AGEs/RAGE interactions acting to prevent the development of diabetic neuropathy or neurodegenerative diseases, could thus be considerable [7]. AMPK is not only a multifunctional metabolic and energy sensor to regulate glucose and lipid homeostasis, its activation also inhibits cell death in several different cells, including neurons [10]. AMPK activation protects against AGEs-mediated oxidative stress in neural cells, as has been documented [24]. Metformin is the drug of choice for treating diabetes via AMPK activation [8]. Due to its capacity to improve hyperglycemia and oxidative stress regulation, metformin protects the nerve from damages related to chronic hyperglycemia [8]. Our data showed that decreased cell viability parallels with increased RAGE protein levels in SH-SY5Y cells after 24 h AGEs stimulation. Moscatilin exhibited a beneficial effect similar to metformin to rescue cell growth and downregulate RAGE expression, but Compound C, an antagonist of AMPK, blocked these actions [25]. The moscatilin inhibition of AGEs/RAGE-induced response in SH-SY5Y cells through AMPK activation could be considerable. We thus elucidate the possible mechanisms of moscatilin by which the activation of AMPK-dependent signaling in SH-SY5Y cells reverses the AGEs-induced damage to prevent cell death.

Intracellular ROS generated in neuronal cells exposed to AGEs is well known [26]. Since AGEs-induced oxidative stress leads to a significant influence in the development of diabetic neuropathy and neurodegenerative disorders, a logical therapeutic approach is to prevent oxidative stress by increasing antioxidant defense [27]. The antioxidant activity of moscatilin has been reported [28]. Therefore, we investigated whether the action of moscatilin on the rescue cell growth was related to the amelioration of AGEs-induced ROS generation. Pretreatment of SH-SY5Y cells with moscatilin inhibited the AGEs-induced ROS generation and lipid peroxidation and abolished the reduction in the GSH/GSSG ratio. Moscatilin counteracts AGEs-triggered oxidative damage, and cell death in SH-SY5Y cells might be contributing to a balance between oxidative stress and the antioxidant defense system. The results support the idea that studies with antioxidants could be one of the strategies available to treat these conditions [27]. More comprehensive studies are required to evaluate the mechanistic role of moscatilin in AGEs-induced oxidative stress and neuronal cellular damage.

AGEs accumulation evokes carbonyl and oxidative stress, which subsequently damages mitochondrial membrane lipids and leads to a collapse of mitochondrial membrane potential and decrease in ATP [29]. The loss of mitochondrial membrane potential increases mitochondrial permeability, leading to mitochondrial swelling and rupture of the outer mitochondrial membrane, followed by the release of cytochrome c from mitochondria into the cytosol to trigger activation of the caspases’ cascade [30]. Caspase-9 and caspase-3 activation are responsible for *apoptosis* execution, leading to DNA fragmentation and, eventually, cell death through cleaving intracellular targets [30]. Besides, the DNA-repairing abilities were abolished after PARP cleavage by caspase-3 [31]. AGEs stimulation turned on the mitochondrial-mediated apoptotic pathway. We found that SH-SY5Y cells exposed to AGEs exhibited a loss of mitochondrial membrane potential with the elevation of ADP/ATP ratio and enhanced cytochrome c release. Meanwhile, cleaved forms of caspase-9, caspase-3, and PARP increased in SH-SY5Y cells following AGEs exposure. These results supported the hypothesis that dysfunctional mitochondria are a central mediator of neuronal apoptosis in diabetes complications [32]. Similar to the actions of metformin, it is shown that moscatilin protected SH-SY5Y cells against AGEs-induced damage by alleviating all of these events, resulting in decreased cell apoptosis and increased cell viability. Our results thus revealed that moscatilin might act through AMPK activation, reversing the apoptotic cascade involving mitochondrial cytochrome *c* release and death caspase activation to produce a neuroprotective effect on AGEs-induced cell injury and apoptosis in SH-SY5Y cells.

The mitochondrial pathway of apoptosis is mainly regulated by the expression of one or more members of the Bcl-2 protein family; among them, Bax and Bcl-2 are a pair of critical regulatory genes with different functions [33]. The p53 protein is activated by DNA damage, which can directly start Bax protein by translocating to the mitochondria, allowing for mitochondrial membrane permeabilization and apoptosis [33]. In the present study, SH-SY5Y cells’ exposure to AGEs decreased Bcl-2 and accompanied by increasing Bax and p53 expression. Moscatilin prevented SH-SY5Y-induced Bcl-2 protein downregulation and also lowered Bax and p53 levels in SH-SY5Y cells; the actions were close to those of metformin. Therefore, moscatilin-induced reductions of the *AGE*-induced *mitochondrial* abnormalities may inhibit neuronal apoptosis, which was associated with restoring the balance between the pro-and anti-apoptotic proteins of the Bcl-2 family.

AGEs/RAGE interactions lead to the activation and translocation of NF-κB, which regulates the expression of genes that control multiple physiology and pathology processes, including cell proliferation and apoptosis, and other essential cell events [34]. AMPK signaling suppresses NF-κB activation has been reported [35]. In the present study, pretreatment with Compound C abated the suppression of moscatilin on NF-κB as well as RAGE expression induced by AGEs, indicating that the inhibition of moscatilin on AGEs caused oxidative stress, and neuronal cellular damage in SH-SY5Y cells was dependent on AMPK activation. The kinase activity of AMPK is supported by the phosphorylation of Thr172 in the α-subunit [36]. We observed a dramatic decrease in the phosphorylation level of AMPK in SH-SY5Y cells exposed to AGEs without an accompanying increase in the total protein level of AMPK. Compound C abolished the revision of moscatilin on the AGEs-induced reduction in AMPK phosphorylation. We propose that moscatilin suppresses AGEs induced RAGE expression by activating AMPK, which reduces intracellular formation ROS and NF-κB activation. These above events finally reverse the apoptotic cascade caused by mitochondrial dysfunction. (Figure 6). Although moscatilin possesses the ability to facilitate tau phosphorylation in an in vitro Alzheimer’s disease-like model developed by cotransfection with the amyloid precursor protein and tau-related plasmids, or is induced by okadaic acid, it cannot evaluate the effect of moscatilin on the mitigation of glycation mediated damages in neuronal cells [37]. Here, we provide evidence that moscatilin alleviates AGEs-induced mitochondrial dysfunction. The mechanism is predicted to occur through modulating AMPK activation and AGEs/RAGE/NF-κB suppression.

In conclusion, the present study results indicate that moscatilin could protect SH-SY5Y cells against AGEs-induced injury and mitochondrial dysfunction; the possible mechanisms involved in *AMPK activation* and *RAGE/NF-κB pathway* suppression. Moscatilin might offer a promising therapeutic strategy to reduce the neurotoxicity or AMPK dysfunction induced by AGEs. It is This will provide a potential benefit for patients with AGE-modification in aging, neurodegenerative diseases, and diabetes.

## 4. Materials and Methods

### 4.1. Cell Culture

The human neuroblastoma SH-SY5Y cell line was obtained from American Type Culture Collection (Manassas, VA). Cells were cultured in growth media composed of Dulbecco’s modified Eagle’s medium (DMEM) containing 10% fetal bovine serum (FBS), 1% penicillin/streptomycin, and 8 mmol/L glucose for 24 h to allow for adherence. Following adherence, growth media was replaced with differentiating media of DMEM/F12 (1:1) medium supplemented with 1% FBS, 8 mmol/L glucose, and 10 μmol/L retinoic acid for five days for neuronal cell phenotype development. Treatments for all experiments took place following the five-day differentiation period.

### 4.2. AGEs Stimulation and Treatments

Cells were seeded at a density of 2 × 10^6^ cells/well in 6-well plates. Upon confluence, cultures were passaged by dissociation in 0.05% (*w*/*v*) trypsin (Sigma-Aldrich, St. Louis, MO, USA) in phosphate-buffered saline (PBS) pH 7.4. For AGEs-induced functional studies, cells were maintained in fresh medium containing 1% FBS for 2 h prior to use in the experiments. Later, cells were pretreated with moscatilin (HongKong PE Biosciences Limited., ShangHai, China, purity ≥ 98%) at different concentrations (0.1, 0.5 or 1.0 μmol/L), or metformin hydrochloride (Abcam, Cambridge, UK; 2 μmol/L) for 1 h followed by exposure to various concentrations of AGEs in bovine serum albumin (AGEs-BSA) (Sigma-Aldrich; 5, 25, 50, and 100 μg/mL) for different timepoints (1, 3, 6, 12, 24 and 48 h) without medium change. The dosage regimen of moscatilin was selected based on a previous report demonstrating that this compound at these concentrations were able to significantly attenuate protect retinal cells from ischemia/hypoxia without cytotoxicity [19]. *Compound C* (Sigma-Aldrich, 5 μmol/L) was added 1 h before moscatilin or metformin were stimulated. Metformin at the indicated concentration on the inhibition of AGEs was served as the positive control group [38]. Metformin was dissolved in water at a concentration of 50 mmol/L and further diluted with culture medium to the final concentrations. Powders of moscatilin or Compound C were dissolved in dimethyl sulfoxide (DMSO, Sigma-Aldrich) to create a 100-μmol/L stock solution, which was subsequently diluted in culture medium to the appropriate concentrations for subsequent experiments. The final DMSO concentration was less than 0.1% (*v*/*v*), a concentration that did not affect cell viability. The following experiments were assessed after treatment. For each experimental repeat, each condition was tested with three wells of cells and each experiment was performed at least five times independently.

### 4.3. Cell Viability Measurements

The cell viability was determined by a c3-(4, 5-dimethylthiazol-2-yl)-2, 5 diphenyl tetrazolium bromide (MTT; Sigma-Aldrich) assay [39]. Briefly, MTT was dissolved in PBS at a concentration of 5 mg/mL. After incubation for 48 h, the MTT solution (25 μL) was added to each well of 96-well plates and incubated at 37 °C in a 5% CO_2_ atmosphere. At the end of the 4 h incubation period, the MTT solution was discarded and 100 μL DMSO was added to all wells to solubilize the *MTT formazan* crystals. The absorbance of the samples was measured at 570 nm on a microplate reader (SpectraMax M5, Molecular Devices, Sunnyvale, CA, USA), and background optical densities were subtracted from that of each well. The optical densities were normalized in percentages relative to the drug-free control (100%).

### 4.4. Intracellular ROS Production

The oxidant-sensing fluorescent probe 2′,7′-dichlorofluorescin diacetate (DCFH-DA) was used to detect intracellular ROS generation [40]. After the treatment period, cells were harvested and incubated in cell-free medium with 10 μmol/L DCFH-DA (Sigma-Aldrich) at 37 °C for 30 min and then washed three times with PBS. After the centrifugation of lysates at 15,000× *g* for 15 min at 4 °C, 50 μL of the supernatant was transferred to a black 96-well plate for fluorescence measurement using a microplate reader (SpectraMax M5) with the excitation at 488 nm and emission at 530 nm.

### 4.5. Lipid Peroxidation Measurement

The thiobarbituric acid reactive substance (TBARS) formed as a by-product of oxidative lipid damage was measured in the cell homogenates by the lipid peroxidation assay kit (Abcam plc., Cambridge, MA, USA) to quantify lipid peroxidation. In brief, 0.5 mL cell lysate (6 × 10^6^ cells/mL) was added to 1 mmol/L ethylene diamine tetraacetic acid (Sigma-Aldrich) and then was mixed with 1 mL cold 15% (*w*/*v*) thiobarbituric acid (TBA) to precipitate proteins. The supernatant mixed with 1 mL of 0.5% (*w*/*v*) TBA in a boiling water bath for 15 min, followed by rapid cooling. The concentration of TBARS in the sample solution was determined from the absorbance at 535 nm and was expressed as nmol/mg protein using a standard solution containing a known concentration of malondialdehyde (MDA) [41]. Total protein concentrations were measured according to the method described in Lowry et al. [42].

### 4.6. Oxidized/Reduced Glutathione Ratio Measurement

The ratios of reduced to oxidized glutathione (GSH/GSSG) were measured with a kit from Cayman (Ann Arbor, MI, USA), which used a spectrophotometric recycling assay to measure the cellular levels of GSH and GSSG. Briefly, cells were scrape-harvested in cold PBS on ice and centrifuged following treatments. Cells were homogenized in cold 2-(*N*-morpholino)ethanesulfonic acid (MES) buffer (0.2 mol/L MES, 50 mmol/L phosphate, 1 mmol/L EDTA, pH 6.0) and centrifuged at 10,000× *g* for 15 min at 4 °C. The supernatants were removed for the assay according to the manufacturer’s instruction. The absorbance was recorded at 405 nm. Data were normalized to total cellular protein content.

### 4.7. Mitochondrial Membrane Potential (MMP) Measurement

The cellular MMP was measured by MMP assay kit containing *5*,*5′*,*6*,*6′-tetrachloro-1*,*1′*,*3*,*3′*- *tetraethylbenzimi*-dazolylcarbocyanine *iodide* (JC-1) dye (Abcam plc.). Cells were seed in a 96-well plate at a density of 1 × 10^4^ cells/well and incubated with 20 µmol/L JC-1 in growth medium at 37 °C for 30 min, hereafter, the cells were collected by centrifugation at 2500 rpm for 5 min, and the pellets were subsequently resuspended in 0.5 mL PBS. The emissions of the aggregate green monomeric form (530 nm) and the aggregate red form (590 nm) were determined using a F-2500 fluorescence spectrophotometer (Hitachi High Technologies America, Inc., Pleasanton, CA, USA). Finally, the red*/*green fluorescence intensity ratio was calculated to determine changes in MMP [43].

### 4.8. ADP and ATP Levels Measurement

The ADP/ATP bioluminescent assay kit from BioAssay Systems (Hayward, CA, USA) is based on luciferase’s ability to produce light in the presence of its substrate luciferin, and ATP was used to measure the cellular ADP and ATP contents [44]. In brief, at the end of treatment, cells were lysed with 10% tricholoroacetic acid, neutralized with 1 mol/L KOH and diluted with 100 mmol/L HEPES buffer (pH 7.4). The first step of the assay involved the luciferase-catalyzed reaction of cellular ATP and D-luciferin, which produced a luminescent signal. Then, ADP was converted into ATP through an enzyme reaction, and the newly formed ATP reacted with D-luciferin. The second light intensity represented the total ADP and ATP content. The calculated ADP/ATP ratio was normalized to the total protein contents in the samples.

### 4.9. Cytochrome C Release Measurement

Cells were seeded at a density of 2.4 × 10^5^ cells/well in 6-well plates and then were scraped and spun twice at 600× *g* for 5 min at the end of the treatment. An ice-cold 1X cytosolic buffer was added into the cell pellet, which was resuspended and incubated on ice for 15 min. Cells were homogenized, and the lysate was spun twice at 800× g for 20 min. The resulting supernatant contained the cytosol, including mitochondria, was centrifuged at 10,000× *g* for 15 min to generate the mitochondria pellet. To obtain the mitochondrial fraction, the mitochondrial pellet was washed and spun with a 1X cytosolic buffer at 10,000× *g* for 10 min and then lysed by adding intact mitochondria buffer followed by incubation on ice for 15 min. The remaining supernatant was centrifuged at 16,000× *g* for 25 min; this centrifuged supernatant was the cytosolic fraction. After isolation of the mitochondria and cytosolic fraction, the cytochrome C ELISA kit (Abcam plc., Cambridge, MA, USA) was used to measure the level of cytochrome c, according to the manufacturer’s instructions. The protein concentration was measured by using a Bio-Rad protein assay.

### 4.10. Apoptotic DNA Fragmentation Analysis

The cell death detection ELISA kit (Roche Molecular Biochemicals, Mannheim, Germany) was used to quantitatively detect the cytoplasmic histone-associated DNA fragments after induced cell death. Briefly, the cells were seeded at a density of 2.4 × 10^5^ cells/well in 6-well plates. At the end of the treatment, cytoplasmic extracts from cells were used as an antigen source in a sandwich ELISA with a primary anti-histone mouse monoclonal antibody-coated to the microtiter plate and a second anti-DNA mouse monoclonal antibody coupled to peroxidase. The amount of peroxidase retained in the immunocomplex was determined photometrically by incubating with 2,2′-azino-di-[3-ethylbenzthiazoline sulfonate] (ABTS) as a substrate for 10 min at 20 °C. The change in color was measured at a wavelength of 405 nm by using a Dynex MRX plate reader controlled through PC software (Revelation, Dynatech Laboratories, CA, USA). The optical density at 405 nm (OD405) reading was then normalized to milligrams of protein used in the assay and presented as an apoptotic DNA fragmentation index.

### 4.11. Nuclear and Cytoplasmic Fractions Preparation

Cells were homogenized in a Western lysis buffer (30 mmol/L Tris-HCl, pH 7.4, 250 mmol/L Na3VO4, 5 mmol/L EDTA, 250 mmol/L sucrose, 1% Triton X-100 with protease inhibitor and phosphatase inhibitor cocktail). The homogenate was centrifuged at 800× *g* for 5 min at 4 °C and supernatant containing the cytosolic extract. The nuclear pellet was resuspended in 25 µL, ice-cold nuclear extraction buffer (20 mmol/L 4-(2-hydroxyethyl)-1-piperazineethanesulfonic acid, 0.4 mmol/L NaCl, 1 mmol/L EDTA, 25% glycerol, protease inhibitors 1X) for 30 min. The extract was centrifuged at 14,000× *g* for 10 min at 4 °C, and supernatants containing nuclear extracts were secured.

### 4.12. Western Blot Analysis

Cytosolic (70 μg total protein) and nuclear (50 μg total protein) extracts were separated on a 10% polyacrylamide gel, subjected to electrophoresis on 10% SDS-PAGE, and transferred to nitrocellulose membranes. The protein concentration was measured using a Bio-Rad protein assay. Nonspecific binding on the nitrocellulose membranes was minimized by blocking for 1 h at room temperature with Tris-buffered saline Tween (TBST) buffer (25 mmol/L Tris-HCl, 150 mmol/L NaCl (pH 7.5), and 0.05% Tween 20) containing 5% (*w*/*v*) nonfat dry milk and then incubated overnight at 4 °C with primary antibodies against RAGE (Santa Cruz Biotechnology, Inc., Santa Cruz, CA, USA; sc-365154), AMPKα (Santa Cruz Biotechnology, Inc., sc-74461), p-AMPKα1/2 (Thr 172)(Santa Cruz Biotechnology, Inc., sc-33524), p53 (Santa Cruz Biotechnology, Inc., SC-126), cleaved caspase-9 (Cell Signaling Technology, Beverly, CA, USA, #9509), cleaved caspase-3 (Cell Signaling Technology, #9661), cleaved poly (ADP-ribose) polymerase-1 (PARP-1) (Santa Cruz Biotechnology, Inc., sc-56196), Bcl-2 (Santa Cruz Biotechnology, Inc., sc-7382), Bax (Santa Cruz Biotechnology, Inc., sc-7480), NF-κB p65 (Santa Cruz Biotechnology, Inc., sc-8008), p-NF-κB p65 (Santa Cruz Biotechnology, Inc., sc-136548). IκBα (Santa Cruz Biotechnology, Inc., sc-1643) and p-IκBα (Ser 32) (Santa Cruz Biotechnology, Inc., sc-8404). The β-actin antibody (Santa Cruz Biotechnology, Inc., sc-47778) was used as an internal control in immunoblotting. The histone H1 antibody (Santa Cruz Biotechnology, Inc., sc-8030) was used as a nuclear loading control. All antibodies were utilized at 1:1000 dilution. After washing with TBST 3 times, the membranes were labeled with horseradish peroxidase-conjugated secondary antibodies for 1 h at room temperature. The blots were washed three times with TBST and incubated with horseradish peroxidase-conjugated secondary antibody for 1 h, and immunoreactive bands were visualized using an enhanced chemiluminescence detection kit (Amersham Biosciences, Buckinghamshire, UK) according to the manufacturer’s instructions. The relative amount of immunoreactive protein in each band was measured by scanning densitometric analysis using the ATTO Densitograph Software (ATTO Corp., Tokyo, Japan) and quantified as the ratio to histone or β-actin. The mean value for samples from the control medium (not treated) on each immunoblot, expressed in densitometry units, was adjusted to a value of 1.0. All experimental sample values were then expressed relative to this adjusted mean value. Cells were sampled from five independent experiments.

### 4.13. Statistical Analysis

Data are expressed as the mean ± standard deviation (SD). Statistically significant differences were evaluated by one-way analysis of variance and Dunnett range post-hoc comparisons using Systat SigmaPlot version 12.5 (Systat Software Inc., San Jose, CA, USA. Differences were considered statistically significant at *p* < 0.05.

## Figures and Tables

**Figure 1 molecules-25-04574-f001:**
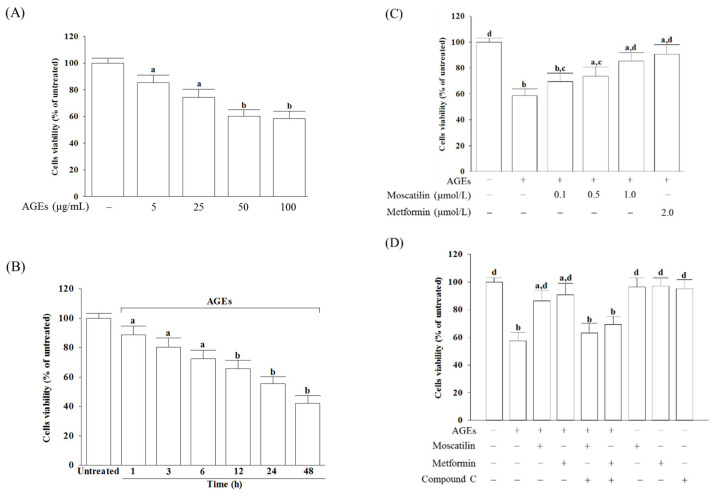
Influences on the decreased cell viability in advanced glycation end-products (AGEs)-treated SH-SY5Y cells. (**A**) SH-SY5Y cells were incubated with various concentrations of AGEs for 24 h. (**B**) SH-SY5Y cells were incubated with 50 μg/mL AGEs for various timepoints. (**C**) SH-SY5Y cells were pretreated with different concentrations of moscatilin (0.1, 0.5 or 1 μmol/L) or metformin (2 μmol/L) for 1 h, followed by exposure to 50 μg/mL AGEs for another 24 h. (**D**) Compound C (5 μmol/L) was added 1 h before moscatilin (1 μmol/L) or metformin (2 μmol/L) were stimulated. Cell viability was determined using an MTT assay and expressed as a percentage of the untreated cells. The results are presented as the mean ± SD of five independent experiments (*n* = 5), each of which was performed in triplicate. ^a^
*p* < 0.05 and ^b^
*p* < 0.01 compared to the data from untreated group (vehicle control). ^c^
*p* < 0.05 and ^d^
*p* < 0.01 when compared to the data from cells cultured under AGEs without any treatment.

**Figure 2 molecules-25-04574-f002:**
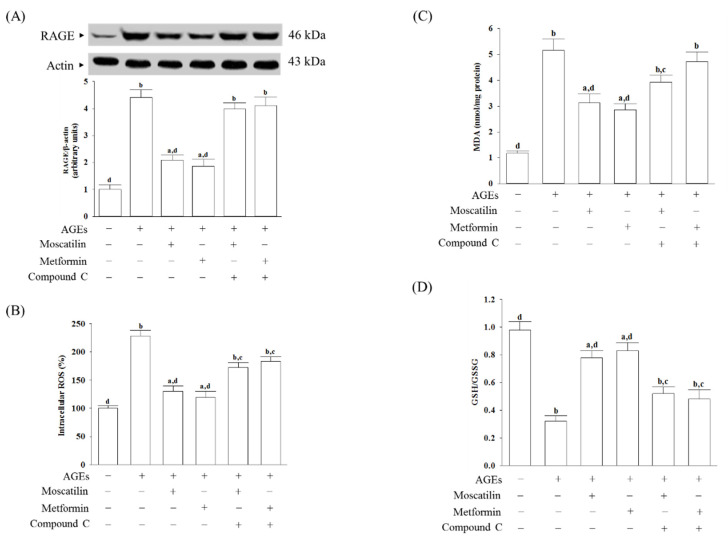
Changes in cell-surface receptors (RAGE) expression and oxidative stress-related factors induced by AGEs in SH-SY5Y cells. Cells were pretreated with moscatilin (1 μmol/L) or metformin (2 μmol/L) for 1 h, followed by exposure to 50 μg/mL AGEs for another 24 h. Compound C (5 μmol/L) was added 1 h before moscatilin or metformin were stimulated. (**A**) The expression of RAGE protein level was determined by Western blot. The densities of protein bands were quantitated and listed in the bottom panels. (**B**) Intracellular ROS production was measured using the oxidation-sensitive fluoroprobe DCFH-DA (**C**) The estimation of accumulated MDA by a thiobarbituric acid test as a measure of lipid peroxidation. (**D**) The cellular GSH-to-GSSG ratio was determined by qualified commercial assay kits. The results are presented as the mean ± SD of five independent experiments (*n* = 5), each of which was performed in triplicate. ^a^
*p* < 0.05 and ^b^
*p* < 0.01 compared to the data from untreated group (vehicle control). ^c^
*p* < 0.05 and ^d^
*p* < 0.01 when compared to the data from cells cultured under AGEs without any treatment.

**Figure 3 molecules-25-04574-f003:**
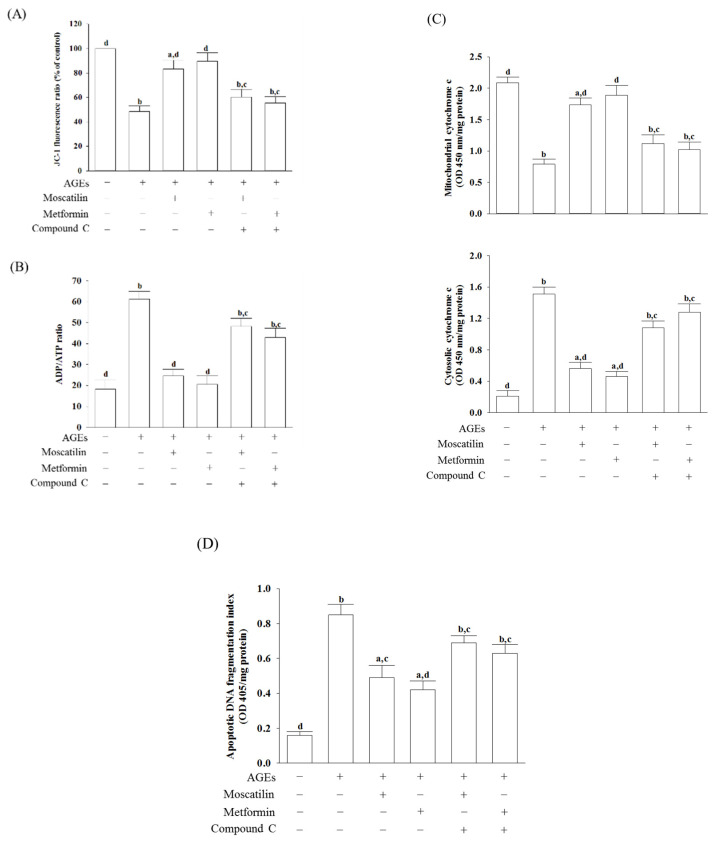
Influences on mitochondrial functions in AGEs-treated SH-SY5Y cells. Cells were pretreated with moscatilin (1 μmol/L) or metformin (2 μmol/L) for 1 h, followed by exposure to 50 μg/mL AGEs for another 24 h. Compound C (5 μmol/L) was added 1 h before moscatilin or metformin were stimulated. (**A**) MMP was assessed using JC-1 fluorescence dye. (**B**) ADP/ATP ratio was measured by commercial assay kit based on the bioluminescent detection of ADP and ATP levels. (**C**) The levels of cytochrome c were determined by immunoassay in both mitochondrial and cytosolic fractions. (**D**) Levels of *DNA fragmentation* were measured by using the cell death detection ELISA Plus kit. The results are presented as the mean ± SD of five independent experiments (*n* = 5), each of which was performed in triplicate. ^a^
*p* < 0.05 and ^b^
*p* < 0.01 compared to the data from untreated group (vehicle control). ^c^
*p* < 0.05 and ^d^
*p* < 0.01 when compared to the data from cells cultured under AGEs without any treatment.

**Figure 4 molecules-25-04574-f004:**
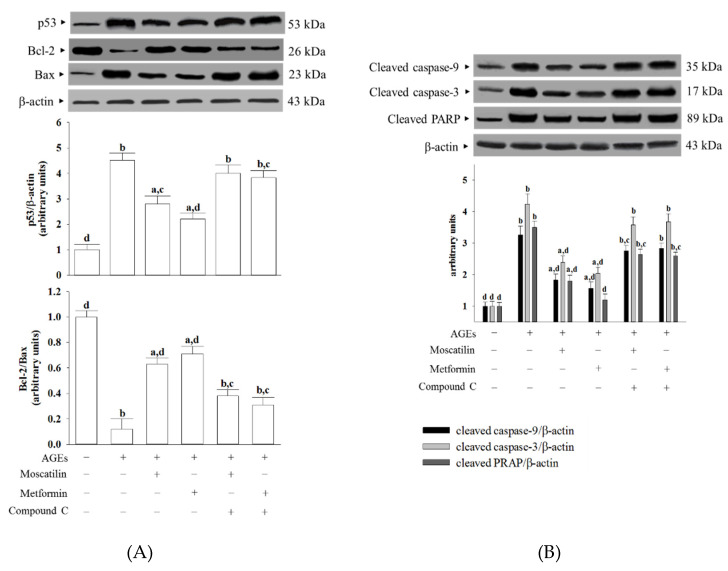
Effects on the pro-apoptotic and anti-apoptotic proteins in AGEs-treated SH-SY5Y cells. Cells were pretreated with moscatilin (1 μmol/L) or metformin (2 μmol/L) for 1 h, followed by exposure to 50 μg/mL AGEs for another 24 h. Compound C (5 μmol/L) was added 1 h before moscatilin or metformin were stimulated. The photographs are representative Western blots for (**A**) p53, Bcl-2 and Bax, (**B**) cleaved caspase-3 and PARP. The densities of protein bands were quantitated and shown in the bottom panels. The results are presented as the mean ± SD of five independent experiments (*n* = 5), each of which was performed in triplicate. ^a^
*p* < 0.05 and ^b^
*p* < 0.01 compared to the data from untreated group (vehicle control). ^c^
*p* < 0.05 and ^d^
*p* < 0.01 when compared to the data from cells cultured under AGEs without any treatment.

**Figure 5 molecules-25-04574-f005:**
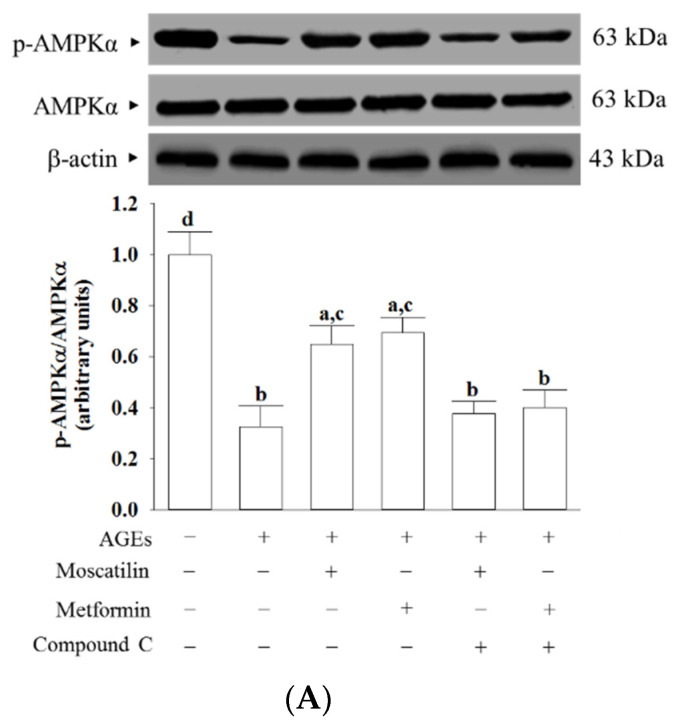
Effects on the AMPK and NF-κB signaling in AGEs-treated SH-SY5Y cells. Cells were pretreated with moscatilin (1 μmol/L) or metformin (2 μmol/L) for 1 h, followed by exposure to 50 μg/mL AGEs for another 24 h. Compound C (5 μmol/L) was added 1 h before moscatilin or metformin were stimulated. The photographs represent the Western blots for (**A**) p-AMPKα and AMPKα, (**B**) p-IκBα and IκBα, (**C**) p-p65 and p65. The densities of protein bands were quantitated and the ratios of p-AMPKα/AMPKα, p-IκBα/IκBα and p-p65/p65 were calculated and shown in the bottom panels. The results are presented as the mean ± SD of five independent experiments (*n* = 5), each of which was performed in triplicate. ^a^
*p* < 0.05 and ^b^
*p* < 0.01 compared to the data from untreated group (vehicle control). ^c^
*p* < 0.05 and ^d^
*p* < 0.01 when compared to the data from cells cultured under AGEs without any treatment.

**Figure 6 molecules-25-04574-f006:**
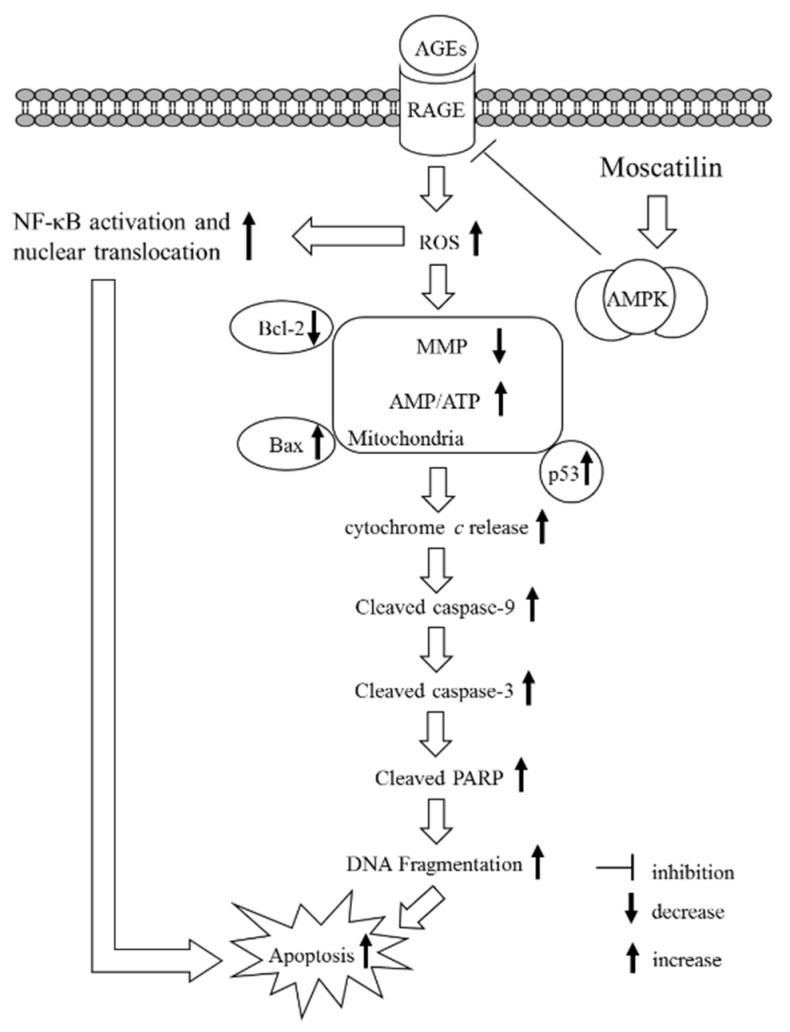
A summary diagram for the possible action of moscatilin on AGEs-treated SH-SY5Y cells. Moscatilin suppresses AGEs-induced RAGE expression by activating AMP-activated protein kinase (AMPK), thereby reducing intracellular formation ROS and NF-κB activation. These events subsequently alleviate mitochondrial dysfunction and finally result in reversing the mitochondrial apoptotic cascades associated with caspase-9 and -3 activation and the cleavage of PARP.

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
