# Peer review of "A Bibenzyl Component Moscatilin Mitigates Glycation-Mediated Damages in an SH-SY5Y Cell Model of Neurodegenerative Diseases through AMPK Activation and RAGE/NF-κB Pathway Suppression"

_molecules, 2020, doi:10.3390/molecules25194574_

Round 1

Reviewer 1 Report

The paper by Lai et al. analyzes the growth inhibition of SH-SY5Y cells by AGE and its reversal by the plant compound moscatillin. The effects of moscatillin are mimicked by metformin and are reduced by CpdC, suggesting that they may be mediated by AMPK. This is a straightforward paper that looks at known effects of AGE, especially generation of ROS, pro-inflammatory stimuli, and apopotosis. The approach uses essentially the same AMPK-related experiments as a previous paper that focused on neuronal stem cells instead of SH-SY5Y cells, so the novelty is low. The experimental outcomes are clear and the statistical analysis is appropriate.

The main problem with the paper (as with the previous paper referred to above) is its use of metformin as a surrogate for AMPK activation. Metformin has multiple effects on metabolism, of which inhibition of mitochondrial respiration is the best documented one. This inhibition causes energy stress, which activates AMPK and many other proteins. Since metformin inhibits oxidative phosphorylation, it also affects ROS levels, which is a main focus of this study, independent of AMPK. Similarly, CpdC is a relatively dirty kinase inhibitor that inhibits many different protein kinases, and is nowadays no longer accepted as a surrogate for AMPK inhibition. Ideally, the authors should validate a few of their key findings using genetic AMPK activation and inhibition, which would generate much more convincing data. At the very least, they should use a direct and more specific small molecule AMPK activator for validation, such as Merck 991 or even Abbot A769662, which are commercially available.

Other points that need to be addressed:
1. It is a far stretch to claim that treatment of a neuroblastoma cell line with an artificial AGE (glycated BSA) is a model for neurodegenerative disease. That statement needs to be toned down.

  1. There is no evidence that metformin activates AMPK through direct binding. The statement that metformin is an AMPK agonist is therefore not correct.
  2. While it is well established that ROS induce strong energy stress, and thereby activate AMPK, conversely evidence that AMPK activation reduces ROS is much less established.
  3. Western blots: please provide size markers and non-cropped blots; this can be done in the form of supplemental figures. It would further be nice to validate that the cross-reactive bands in the displayed narrow gel slices are indeed the proposed proteins.
  4. In Fig. 3B, the authors show that metformin counters energy stress induced by AGE in SH-SY5Y cells. That is very surprising given metformin’s well-established effect of generating energy stress in cells. How do the authors explain this surprising result? Ideally, the authors would also test the effect of metformin in the absence of AMPK or in the presence of an AMPK mutant that is non-responsive to changes in adenine nucleotide levels – that may be too much work for this paper, but the authors should at least discuss the unexpected relief of energy stress by metformin.
  5. Fig. 5A: the top gel slice is not labeled.
  6. Please exchange “theory” ‘that dysfunctional mitochondria are central mediators of neuronal apoptosis in diabetic complications’ against “hypothesis”.
  7. There are a number of statements in the paper that do not convey a real message, e.g., “…pathology associated with hyperglycemia ..involves chronic hyperglycemia” and others. That is most likely just a language issue as there are also many grammar issues. I would therefore recommend to have the manuscript copy-edited, ideally by a native speaker.

Author Response

Ms. Ref. No.:  molecules-933122 R1
Title: A Bibenzyl Component Moscatilin Mitigates Glycation Mediated Damages in an SH-SY5Y Cell Model of Neurodegenerative Diseases through AMPK Activation and RAGE/NF-?B Pathway Suppression

Authors: Lai MC et al.

Dear distinguished referee:

Thank you very much for reading this manuscript and the helpful comments. The revision has been amended according to your kind suggestions as follows:

Comments and Suggestions for Authors

  1. The paper by Lai et al. analyzes the growth inhibition of SH-SY5Y cells by AGE and its reversal by the plant compound moscatillin. The effects of moscatillin are mimicked by metformin and are reduced by CpdC, suggesting that they may be mediated by AMPK. This is a straightforward paper that looks at known effects of AGE, especially generation of ROS, pro-inflammatory stimuli, and apopotosis. The approach uses essentially the same AMPK-related experiments as a previous paper that focused on neuronal stem cells instead of SH-SY5Y cells, so the novelty is low. The experimental outcomes are clear and the statistical analysis is appropriate.

Reply

To study the effects and mechanism of moscatillin, one of the active compounds from Dendrobium species, on the glycation-induced neuron dysfunction, an in vitro cell model established with AGEs-treated SH-SY5Y cells are performed according to the several previous studies (Int J Mol Med 2018;41(5):2855-2864; Acta Pharmacol Sin 2011;32(8):991-8). The in vitro cell model and the experimental methods we used are all known and established, without considering the novelty. We hope you can understand it.

  1. The main problem with the paper (as with the previous paper referred to above) is its use of metformin as a surrogate for AMPK activation. Metformin has multiple effects on metabolism, of which inhibition of mitochondrial respiration is the best documented one. This inhibition causes energy stress, which activates AMPK and many other proteins. Since metformin inhibits oxidative phosphorylation, it also affects ROS levels, which is a main focus of this study, independent of AMPK. Similarly, CpdC is a relatively dirty kinase inhibitor that inhibits many different protein kinases, and is nowadays no longer accepted as a surrogate for AMPK inhibition. Ideally, the authors should validate a few of their key findings using genetic AMPK activation and inhibition, which would generate much more convincing data. At the very least, they should use a direct and more specific small molecule AMPK activator for validation, such as Merck 991 or even Abbot A769662, which are commercially available.

Reply

Metformin has been shown to act via AMPK-dependent and AMPK-independent mechanisms. In contrast, increasing shreds of evidence have shown that metformin inhibits AGEs-induced actions that might involve in the regulation of AMPK. For example, metformin inhibits AGEs-Induced inflammatory response in murine macrophages through AMPK activation has been documented (J Diabetes Res 2016;2016:4847812). Besides, metformin could inhibit the AGEs-induced apoptosis and inflammatory and fibrotic reactions in tubular cells, probably by reducing ROS generation via suppression of RAGE expression through AMPK activation (Horm Metab Res 2012;44:891-895). Our research refers to the articles that indicate the neuroprotective role of metformin in advanced glycation end product treated human neural stem cells is AMPK-dependent (Biochim Biophys Acta 2015;1852:720-731). Compound C is occasionally used as an AMPK inhibitor but also inhibits several other kinases. In the present study, Compound C abolished the revision of moscatilin on the AGEs-induced reduction in AMPK phosphorylation (Fig 5A); thus, we predicted that moscatilin alleviates AGEs-induced SH-SY5Y cells damage is to be probably through modulating AMPK activation. The manuscript should reverse in the time constraint given. However, we will use the direct and more specific AMPK activator or AMPK inhibitor to further confirm the effects and mechanism of moscatillin on the mitigation of glycation-induced neuron dysfunction according to your kind instructions. We wish this interpretation would be satisfactory.

Other points that need to be addressed:

  1. It is a far stretch to claim that treatment of a neuroblastoma cell line with an artificial AGE (glycated BSA) is a model for neurodegenerative disease. That statement needs to be toned down.

Reply

The statement on AGEs-induced injury in SH-SY5Y cells (line 4 of the 4th paragraph on Page 2) has been revised according to your instructions. We hope you consider this change sufficient and acceptable.

  1. There is no evidence that metformin activates AMPK through direct binding. The statement that metformin is an AMPK agonist is therefore not correct.

Reply

The indicated statement (line 2 of the 2nd paragraph on Page 2) has been corrected to metformin is an AMPK activator instead of an AMPK agonist. We hope you consider this change sufficient and acceptable.

  1. While it is well established that ROS induce strong energy stress, and thereby activate AMPK, conversely evidence that AMPK activation reduces ROS is much less established.

Reply

Metformin reverses AGE-induced oxidative stress in human neural stem cells via AMPK-dependent downregulation of RAGE levels has been clarified (Exp Cell Res 2017;359:367-373). Metformin can also inhibit the AGEs-induced apoptosis and inflammatory and fibrotic reactions in tubular cells by reducing ROS generation via suppression of RAGE expression through AMPK protein kinase activation (Horm Metab Res 2012;44:891-895). Not only metformin, but it has also been demonstrated that the total polyphenol of Anemarrhena asphodeloides inhibited AGEs-induced ROS-associated inflammation and ameliorated endothelial dysfunction through beneficial regulation of AMPK activation (J Diabetes. 2014;6:304-315). AMPK activation protects cells against AGEs induced injury by blocking the AGEs-RAGE-ROS axis has been considerable. However, the mechanism between AMPK activation and AGEs-RAGE-ROS axis blockade needs further evaluation in the future. We wish this interpretation would be satisfactory and acceptable.

  1. Western blots: please provide size markers and non-cropped blots; this can be done in the form of supplemental figures. It would further be nice to validate that the cross-reactive bands in the displayed narrow gel slices are indeed the proposed proteins.

Reply

The raw data (a computer file) of uncropped blots was lost due to a computer virus attack; we hope you can understand it. However, the molecular weight of the detected protein was labeled in the figure. We honestly wish it would be suitable to meet your requirements.

  1. In Fig. 3B, the authors show that metformin counters energy stress induced by AGE in SH-SY5Y cells. That is very surprising given metformin’s well-established effect of generating energy stress in cells. How do the authors explain this surprising result? Ideally, the authors would also test the effect of metformin in the absence of AMPK or in the presence of an AMPK mutant that is non-responsive to changes in adenine nucleotide levels – that may be too much work for this paper, but the authors should at least discuss the unexpected relief of energy stress by metformin.

Reply

We summarized our study results indicated that moscatilin suppresses AGEs induced RAGE expression by activating AMPK, which leads to block the binding of AGEs to RAGE, thereby reduce intracellular formation ROS and NF-κB activationThese above events finally result in reversing the apoptotic cascade caused by mitochondrial dysfunction (line 11-15 of the last paragraph on Page 9). A summary diagram covering our findings was shown in Figure 6 (Page 10). We hope you consider this improvement sufficient and acceptable.

  1. 5A: the top gel slice is not labeled.

Reply

Fig. 5A has been corrected. We hope you consider this improvement sufficient and acceptable.

  1. Please exchange “theory” ‘that dysfunctional mitochondria are central mediators of neuronal apoptosis in diabetic complications’ against “hypothesis”.

Reply

The word “theory” has been replace by “hypothesis” (line 13 of the 2nd paragraph on Page 9) according to your instructions. We hope you consider this improvement.

  1. There are a number of statements in the paper that do not convey a real message, e.g., “…pathology associated with hyperglycemia ..involves chronic hyperglycemia” and others. That is most likely just a language issue as there are also many grammar issues. I would therefore recommend to have the manuscript copy-edited, ideally by a native speaker.

Reply

The indicated sentence has been revised according to your recommendation. Please find it in the first 3 lines of the first paragraph on Page 2. We highly appreciate your helpful comments. We regret there were problems with the English. The paper has been carefully revised by a native English speaker to improve the grammar and readability. We honestly wish it would be suitable to meet your requirements.

The changes in the revision are highlighted in red. We hope that this revised version of our work will meet your high standards for acceptance. Also, I wish to express my warmest thanks to you again. Your kind agreement of recognition will be sincerely appreciated.

Reviewer 2 Report

Lai et al. in their manuscript entitled “A Bibenzyl Component Moscatilin Mitigates Glycation Mediated Damages in an SH-SY5Y Cell Model of Neurodegenerative Diseases through AMPK Activation and RAGE/NF-kB Pathway Suppression”, have investigated the beneficial effects of a natural compound moscatilin in an in vitro model mimicking diabetic/hyperglycemia-associated neurodegenerative diseases.

Overall, this study was well designed with experiments effectively executed. The methods employed were appropriate and results properly interpreted. This manuscript adds new knowledge to diabetic-associated neurodegenerative disease literature and is of interests to the readers of this journal. However, the reviewer does have some concerns as outlined below.

Major concerns:

1. Is AGEs-induced injury in SH-SY5Y cells a general model for neurodegenerative diseases or a specific one for diabetic/hyperglycemia-associated neurodegenerative diseases? If it is the latter, then it should explicitly state so in the Abstract and in the Introduction such as the last sentence of the first paragraph.

2. The authors showed AGEs treatment induced RAGE protein levels – Figure 2A. RAGE is the cell surface receptor for AGEs and it is interesting to know whether this feed-forward stimulation the authors observed occurs at the transcription level or the translation level. A real-time PCR experiment can reveal the expression changes of RAGE at the mRNA levels and it will complement the immunoblotting data.

3. The authors have showed an array of changes when the SH-SY5Y cells were treated with AGEs. It would be a nice addition if the authors can prepare a summary diagram covering the findings in the Results section.

4. The authors need to clarify how moscatilin-induced AMPK activation fits in  its beneficial effects against AGEs-induced injury in SH-SY5Y cells. For example, the authors stated in the Discussion section (2nd paragraph on page 9) that “… it is shown that moscatilin protected SH-SY5Y cells against AGEs-induced damage by ameliorating all of these events, resulting in decreased cell apoptosis and increased cell viability. Our results thus revealed that moscatilin might act through reversing the apoptotic cascade involving mitochondrial cytochrome c release and death caspase activation to produce a neuroprotective effect on AGEs induced cell injury and apoptosis in SH-SY5Y cells.” If moscatilin can do all the above, then where does AMPK activation fit in?

Minor concerns:

The manuscript was overall well written, except there were grammatical errors that disrupt the logic flow and cause confusion during reading. A thorough check would greatly improve the readability and facilitate navigation of the paper. Just name a few:

  1. The first sentence in Introduction.
  2. Line 5 in page 2.
  3. The last sentence of the first paragraph on page 2.
  4. The first sentence in the 4th paragraph on page 2.
  5. The first sentence in the 2nd paragraph on page 4

Some typos:

  1. In the Abstract, “mode” should be “model”
  2. In Results 2.1, in the text there was ug/mL, but in the figure, it was umol/mL.

Author Response

Ms. Ref. No.:  molecules-933122 R1
Title: A Bibenzyl Component Moscatilin Mitigates Glycation Mediated Damages in an SH-SY5Y Cell Model of Neurodegenerative Diseases through AMPK Activation and RAGE/NF-?B Pathway Suppression

Authors: Lai MC et al.

Dear distinguished referee:

Thank you very much for reading this manuscript and the helpful comments. The revision has been amended according to your kind suggestions as follows:

Major concerns:

  1. Is AGEs-induced injury in SH-SY5Y cells a general model for neurodegenerative diseases or a specific one for diabetic/hyperglycemia-associated neurodegenerative diseases? If it is the latter, then it should explicitly state so in the Abstract and in the Introduction such as the last sentence of the first paragraph.

Reply

The statement on AGEs-induced injury in SH-SY5Y cells (Abstract and line 4 of the 4th paragraph on Page 2) on has been revised according to your instructions. We hope you consider this change sufficient and acceptable.

  1. The authors showed AGEs treatment induced RAGE protein levels – Figure 2A. RAGE is the cell surface receptor for AGEs and it is interesting to know whether this feed-forward stimulation the authors observed occurs at the transcription level or the translation level. A real-time PCR experiment can reveal the expression changes of RAGE at the mRNA levels and it will complement the immunoblotting data.

Reply

Similar to the previous study that indicated the RAGE protein levels were significantly increased in hNSCs treated with AGEs (Exp Cell Res. 2017;359(2):367-373), we observed that AGEs induced RAGE protein levels in SH-SY5Y cells (Figure 2A). DM patients with microvascular and macrovascular complications showed a 12 fold and 8 fold higher RAGE mRNA expression than healthy controls; circulating AGEs level showed a significant positive correlation with RAGE mRNA expression could thus be considerable (Microvasc Res. 2014;95:1-6). The manuscript should reverse in the time constraint given. However, a real-time PCR experiment will perform to reveal the expression changes of RAGE at the mRNA levels in the future according to your suggestion; we hope you can understand it.  

  1. The authors have showed an array of changes when the SH-SY5Y cells were treated with AGEs. It would be a nice addition if the authors can prepare a summary diagram covering the findings in the Results section.

Reply

Our study results are summarized in the line 11-15 of the last paragraph on Page 9. A summary diagram covering our findings was shown in Figure 6 (Page 10). We hope you consider this improvement sufficient and acceptable.

  1. The authors need to clarify how moscatilin-induced AMPK activation fits in  its beneficial effects against AGEs-induced injury in SH-SY5Y cells. For example, the authors stated in the Discussion section (2ndparagraph on page 9) that “… it is shown that moscatilin protected SH-SY5Y cells against AGEs-induced damage by ameliorating all of these events, resulting in decreased cell apoptosis and increased cell viability. Our results thus revealed that moscatilin might act through reversing the apoptotic cascade involving mitochondrial cytochrome c release and death caspase activation to produce a neuroprotective effect on AGEs induced cell injury and apoptosis in SH-SY5Y cells.” If moscatilin can do all the above, then where does AMPK activation fit in?

Reply

The indicated sentences has been revised according to your instructions. Please find it in the line 2 on Page 10. We hope you consider this change sufficient and acceptable.

Minor concerns:

The manuscript was overall well written, except there were grammatical errors that disrupt the logic flow and cause confusion during reading. A thorough check would greatly improve the readability and facilitate navigation of the paper. Just name a few:

  1. The first sentence in Introduction.

Reply

The first sentence in Introduction was rewritten according to your instructions. We hope you consider this change sufficient and acceptable.

  1. Line 5 in page 2.

Reply

The indicated sentence has been revised. We hope you consider this change sufficient and acceptable.

  1. The last sentence of the first paragraph on page 2.

Reply

The indicated sentence has been revised. We hope you consider this improvement sufficient and acceptable.

  1. The first sentence in the 4thparagraph on page 2.

Reply

The indicated sentence has been revised. We hope you consider this improvement sufficient and acceptable.

  1. The first sentence in the 2ndparagraph on page 4

Reply

The indicated sentence has been revised. We hope you consider this improvement sufficient and acceptable.

Some tyos:

  1. In the Abstract, “mode” should be “model”

Reply

The typing error has been corrected. Thank you for your guidance.

  1. In Results 2.1, in the text there was ug/mL, but in the figure, it was umol/mL.

Reply

Figure 1A has been corrected. Thank you for your guidance.

The changes in the revision are highlighted in red. We hope that this revised version of our work will meet your high standards for acceptance. Also, I wish to express my warmest thanks to you again. Your kind agreement of recognition will be sincerely appreciated.

Round 2

Reviewer 1 Report

In their revised manuscript, the authors have changed wordings as suggested. However, they have not addressed my main concern, that is the low specificities of metformin and CpdC for AMPK activation and inhibition, respectively. The low specificity of metformin is a particular concern as metformin is known to affect ROS levels independent of AMPK activation. Similarly, CpdC is generally no longer accepted as a surrogate for AMPK inhibition as it also inhibits many other protein kinases. The minimal (and only) experiment requested by this reviewer, use of a commercially available small molecule direct activator to validate a few key experiments, was not pursued. I do not understand the authors response to this request “The manuscript should reverse in the time constraint given”. While both metformin and CpdC cannot be used as evidence that effects are mediated by AMPK, their combination makes a stronger point. Still, I cannot recommend acceptance of this manuscript unless the authors at least acknowledge the limitations of these two compounds.

The response to the request to provide uncropped Western blots (“lost due a computer virus attack”) was also not what a reviewer likes to hear. Without uncropped gels, physical size markers, and antibody validations, it is a lot harder to evaluate the results.

Additional points:

While the summary cartoon is a welcome addition, there are problems with the new summary sentence “…moscatilin suppresses AGEs induced RAGE expression by activating AMPK, which leads to block the binding of AGEs to RAGE, thereby reduce intracellular formation ROS and NF-kB activation”. Without clear evidence, this should not be presented as a fact. Please revise using a phrase such as “We propose that”, “Our data suggest”, or similar. Moreover, the authors have not shown that AMPK blocks the binding of AGEs to RAGEs. Either they present this data (or a reference to previously published data) or they have to delete this statement.  

The English still needs to be corrected, especially the grammar. I understand that this is not easy for non-native speakers, and perhaps the journal editors can help copy-editing.
